# Microstructure and Solute Concentration Analysis of Epitaxial Growth during Wire and Arc Additive Manufacturing of Aluminum Alloy

Ruwei Geng [1,*] , Yanhai Cheng [1], Luqiang Chao [2], Zhengying Wei [3] and Ninshu Ma [4]

1. School of Mechanical Engineering, China University of Mining and Technology, Xuzhou 221116, China
2. Shandong Jiaotong College Mechanical Equipment Technology Company, Jinan 250031, China
3. State Key Laboratory for Manufacturing System Engineering, Xi'an Jiaotong University, Xi'an 710049, China
4. Joining and Welding Research Institute, Osaka University, Mihogaoka, Ibaraki 567-0047, Osaka, Japan
* Correspondence: geng6294@cumt.edu.cn

**Abstract:** Microstructure and solute distribution have a significant impact on the mechanical properties of wire and arc additive manufacturing (WAAM) deposits. In this study, a multiscale model, consisting of a macroscopic finite element (FE) model and a microscopic phase field (PF) model, was used to predict the 2319 Al alloy microstructure evolution with epitaxial growth. Temperature fields, and the corresponding temperature gradient under the selected process parameters, were calculated by the FE model. Based on the results of macroscopic thermal simulation on the WAAM process, a PF model with a misorientation angle was employed to simulate the microstructure and competitive behaviors under the effect of epitaxial growth of grains. The dendrites with high misorientation angles experienced competitive growth and tended to be eliminated in the solidification process. The inclined dendrites are commonly hindered by other grains in front of the dendrite tip. Moreover, the solute enrichment near the solid/liquid interface reduced the driving force of solidification. The inclined angle of dendrites increased with the misorientation angle, and the solute distributions near the interface had similar patterns, but various concentrations, with different misorientation angles. Finally, metallographic experiments were conducted on the WAAM specimen to validate the morphology and size of the dendrites, and electron backscattered diffraction was used to indicate the preferred orientation of grains near the fusion line, proving the existence of epitaxial growth.

**Keywords:** WAAM; microstructure; epitaxial growth; concentration distribution; phase field

## 1. Introduction

Wire and arc additive manufacturing (WAAM) [1] is a high-efficiency, low-cost [2–4] additive manufacturing technology that has significant advantages in fabricating large-scale metallic components. The mechanical properties of WAAM components are greatly determined by their microstructure and solute distributions. Investigating the microstructure evolution and solute concentration is an essential way to actively realize performance control. A large number of quantitative studies have been conducted on the grain structure evolution of metal additive manufacturing by simulations and experiments [5–9]. Wang et al. [8] investigated the heat transfer and dendrite growth in the laser molten pool of Al alloy through a macromicro model, and the simulated grain morphology and primary dendrite arm spacing were in good agreement with experimental observations. Su et al. [9] investigated the effects of process parameters on the microstructure of Al-Mg alloy fabricated by WAAM, realizing the modification of large dendrites to refined equiaxed grains by changing the heat input.

Multiscale simulation is an effective way to predict the solidification structure in metal additive manufacturing, in which computational fluid dynamics (CFD), or the FEM method, is usually employed to compute the macroscopic thermal information served as

the input for the microscopic PF model. Xiao et al. [10] developed a CFD-PF multiscale model to simulate the dendrite growth behavior for laser additive manufacturing. The changing curves of the temperature gradient and solidification speed with different process parameters were obtained, and then, the microstructure evolution was reproduced by the PF model. Zheng et al. [11] represented the interface instability of epitaxial solidification in a weld pool by a multiscale model that integrated the FEM and PF method, and found that the stability of the planar interface increased with the tilting growth angle, while the dendrite spacing was irrelevant to the tilting growth angles.

Since solidification nucleates from the base metal, the preferred orientation of newly formed grains is determined by the grains of the substrate material, which is called epitaxial growth [12,13]. Once the preferred orientation is not parallel to the grain growth direction, fierce competitive behavior among grains occurs. Competition is important for the solidification process, which has significant impacts on the morphology of the microstructure and the ultimate material properties. Chen et al. [14] simulated epitaxial nucleation and competitive growth via a cellular automata model. The temperature gradient was the main factor determining competitive growth. Finally, the growth direction of dendrites tended to be perpendicular to the fusion line. Geng et al. [15] reproduced the solidification morphology evolution during the cellular-to-dendritic transition, and multiple grains with various orientations grew in the competitive stage during the solidification of laser welding. Yang et al. [16] used a 3D PF model to simulate grain evolution with epitaxial growth in powder-bed-fusion additive manufacturing. Grain growth during multilayer deposition was predicted, and a small number of grains formed by continuous epitaxial growth could grow through several deposition layers, which might lead to the formation of texture. However, the composition diffusion between the solid and liquid phases was not considered. V. Pavan Laxmipathy et al. [17] investigated grain competitive growth in the presence of convection, and found that conventional overgrowth behavior translated into an anomalous overgrowth phenomenon, where unfavorably oriented dendrites overgrew at the expense of favorably oriented dendrites. Generally, the morphology evolution and epitaxial growth in the metal additive manufacturing solidification process have been studied extensively. However, the concentration distribution during dendrite epitaxial growth, and the influences of concentration redistribution on competitive behavior, are relatively limited.

In this paper, an FE model was developed to calculate the thermal field and solidification parameters, and served as input for obtaining the microstructure. Subsequently, after taking the epitaxial grain growth into consideration, a microscopic PF model was employed to reproduce the microstructure evolution, as well as solute concentration in the presence of misorientation angles. The influences of the misorientation angle on the dendrite morphology and concentration distribution were determined by PF simulations. Additionally, to verify the simulated results, experimental observations were carried out on the WAAM specimen of 2319 aluminum alloy to obtain the grain morphology and crystal orientation of the grains near the fusion line, by scanning electron microscopy (SEM) and electron backscattered diffraction (EBSD) tests.

## 2. Experimental Procedure and Multiscale Simulation Modelling

### 2.1. Experimental Setup

A variable polarity gas tungsten arc welding system was employed with a Fronius Magic Wave 3000 welding power source (Fronius International GmbH, Pettenbach, Austria). A square wave AC mode was used in the VP-GTA welding process. The plus frequency was set to 5 Hz. Pure argon (99%) was employed as a shielding gas, with flow rates of 15 L/min. The wire material used in this research was ER2319Al alloy (from Alcotec Company, Traverse City, MI, USA), and 2319 Al alloy for the substrate. The chemical composition of the alloy is shown in Table 1.

**Table 1.** The chemical compositions (wt.%) of the substrate and wires.

| Material | Cu | Mg | Mn | Si | Zn | Ti | Al |
|----------|------|------|-----------|------|------|-----------|------|
| ER2319 | 5.8–6.8 | 0.02 | 0.20–0.40 | 0.20 | 0.10 | 0.10–0.20 | *bal.* |

Samples with WAAM-fabricated single deposition layers were selected for metallographic examination, to verify the PF simulation results. The samples were ground using SiC paper, with grits from #200 to #1200, polished with 0.5 μm alumina slurries, and then etched for 15 s in a Keller's etchant (1 mL HF, 1.5 mL HCl, 2.5 mL HNO3, and 90 mL distilled water). The microstructure of the samples deposited with varied WAAM process parameters was evaluated using a scanning electron microscope (SEM). For the complementary EBSD analysis, the test samples were re-polished, using a diamond slurry suspension and a 0.05 μm colloidal silica solution.

## 2.2. Macroscopic FE Model

The microstructure of WAAM is determined by the thermal distributions and evolution in the solidification process. Therefore, a 3D FE model of WAAM for 2319 aluminum alloy was established to calculate the temperature field in the melting pool. The sizes of substrate and deposition layer were shown in Figure 1. The material properties of the alloy used in the FE model were calculated by Jmat-pro software, and nonuniform meshing was employed to maintain a balance between the calculation efficiency and accuracy. The minimum mesh size was 0.125 mm near the deposition zone, and the mesh sizes increased gradually to 0.75 mm in the substrate far away from the deposition layer, as shown in Figure 1.

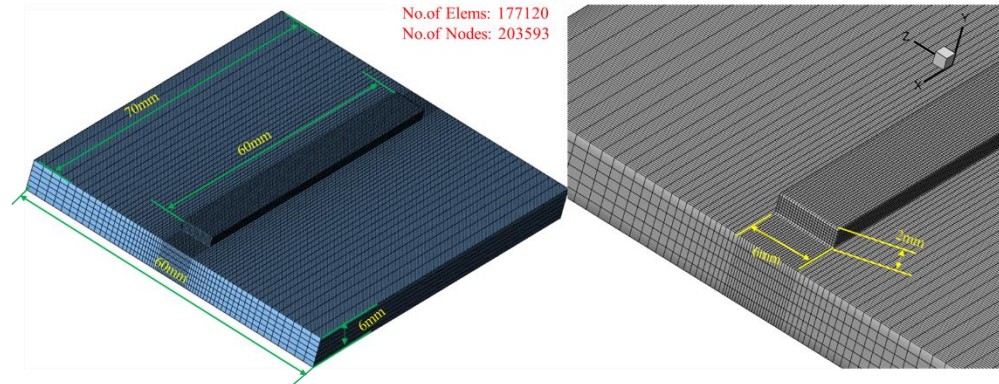

**Figure 1.** Nonuniform meshing of FE model.

The element birth-and-death method was used to simulate the deposition process in additive manufacturing. The initial temperature of the system was set as room temperature, 25 °C. The thermal losses resulting from convection and radiation were taken into consideration in the FE simulations.

A double ellipsoid heat source was used here to calculate the distribution of the instantaneous temperature field [18]. The in-house software, Joining and Welding Research Institute Analysis (JWRIAN) [19,20], developed by JWRI, was used for the FE simulations presented here.

## 2.3. Microscopic Phase Field Model

To reproduce the microstructure evolution microscopically, the quantitative PF model established by J. C. Ramirez [21] and Echebarria [22] was adopted. The PF model can incorporate other physical fields, such as temperature, concentration, and fluid flow. Therefore, PF can precisely predict solid and liquid interface migration, grain competitive growth,

and solute enrichment, which may lead to local constitutional supercooling and further affect grain evolution. The thermal-physical properties used in the

PF simulations are summarized in Table 2. According to the frozen temperature approximation, the transient temperature in the micro domain at the molten pool can be defined as [22]:

$$T(z, t) = T_0 + G(t)(Z - Z_0 - \int_0^t v_p(t')dt') \tag{1}$$

where $T$ is the temperature, $T_0$ is a reference temperature, $G$ is the temperature gradient, $Z$ is the axis along the columnar dendrite growth direction, and $v_p$ is the solidification speed. In the WAAM, $v_p$ and $G$ are calculated from the macroscopic FE model. The phase field model is defined as [23]:

$$\tau_0[1 - (1 - k)\frac{Z - \int_0^t v_p(t')dt'}{l_T}]\frac{\partial \phi}{\partial t} = W^2 \nabla^2 \phi + \phi - \phi^3 - \lambda g'(\phi)(U + \frac{Z - \int_0^t v_p(t')dt'}{l_T}) \tag{2}$$

$$\left(\frac{1+k}{2} - \frac{1-k}{2}\phi\right)\frac{\partial U}{\partial t} = \nabla\left(Dq(\phi)\nabla U + \frac{1}{2\sqrt{2}}[1 + (1 - k)U] \times \frac{\partial \phi}{\partial t}\frac{\nabla \phi}{|\nabla \phi|}\right) \\ + [1 + (1 - k)U]\frac{1}{2}\frac{\partial \phi}{\partial t} \tag{3}$$

where $l_T = |m|(1 - k)c_0/kG$, $\lambda$ describes the strength of the coupling between the phase field and the temperature field, $m$ is the liquidus slope of the alloy, $k$ is the equilibrium partition coefficient, $\tau$ is the relaxation time, $D$ is the solute diffusivity, and $\Phi$ is the order parameter of the phase field. $U$ is the dimensionless concentration, defined as:

$$\mathbf{U} = \frac{\exp(\mu) - 1}{1 - k} \tag{4}$$

$$\mu = \ln(\frac{2c/c_\infty}{1 + k - (1 - k)\phi}) \tag{5}$$

where $c$ is the solute concentration of the Al-6.3 wt.% Cu alloy, and $c_0$ is the initial concentration. $c_\infty$ is the concentration far from the solid-liquid interface, and $W$ is the interface width.

In the WAAM solidification process, cellular dendrites with different misorientation angles can be observed. When the anisotropy of the surface tension and misorientation angle were taken into consideration, the interface width is [24]:

$$W = W_0 a_s(n) = W_0[1 + \gamma \cos 4(\theta - \theta_0)] \tag{6}$$

where $\theta$ is the angle between the interface normal and the axis z, $\theta_0$ is the misorientation angle, which is defined as the angle between the crystallographic preferred orientation and the temperature gradient direction, and $\gamma$ is the anisotropy strength. $W_0$ is taken as 0.27 μm, dx = 0.8 $W_0$, and $\Delta t \leq \frac{(dx)^2}{4D}$ in this paper.

**Table 2.** Thermo-physical properties of the ER2319 alloy.

| Properties | Value |
|---|---|
| Liquidus temperature (K) | 917 |
| Solidus temperature (K) | 821 |
| Latent heat of fusion, $L$ (J·kg$^{-1}$) | $3.89 \times 10^5$ |
| Specific heat capacity, $Cp$ (J·kg$^{-1}$·K$^{-1}$) | 786 |
| Solute diffusivity in the liquid, $D$ (m$^2$·s$^{-1}$) | $3 \times 10^{-9}$ |
| Equilibrium partition coefficient, $k$ | 0.15 |
| Anisotropy, $\gamma$ | 0.02 |

**Table 2.** *Cont.*

| Properties | Value |
|---|---|
| Liquidus slope, $m$ (K·wt%$^{-1}$) | $-2.6$ |
| Initial concentration, $c_0$ (wt%) | 6.3 |
| Interface width, $W_0$ (μm) | 0.27 |
| strength of the coupling between the phase field and the temperature field, $\lambda$ | 10 |
| Gibbs-Thomson coefficient, $\Gamma$ (K·m) | $2.4 \times 10^{-7}$ |

To investigate the epitaxial growth behaviors, the initial conditions were set as in Figure 2. A thin solid layer (20 grid thicknesses) was located at the bottom of the calculation domain, which was divided into three parts. The misorientation angle of the left and right zones (colored in green) was 0, which means that the preferred orientation of grains in these two zones was parallel to the thermal gradient. In the middle part (light yellow color) of the thin solid layer, the misorientation varied on different occasions. Zero Neumann and periodic conditions were employed in the horizontal and vertical boundaries.

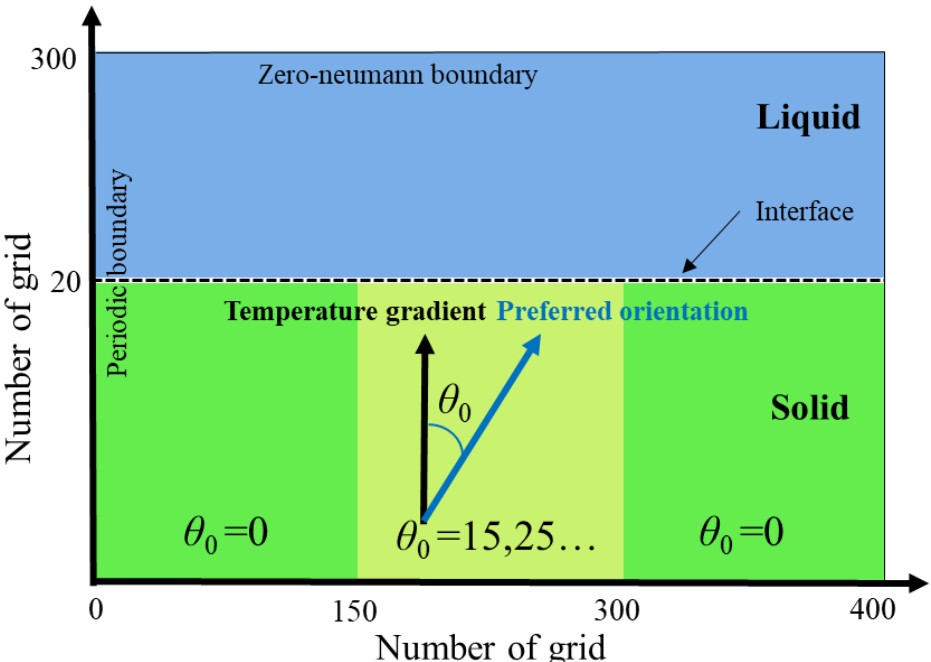

**Figure 2.** Initial conditions of PF model with epitaxial growth.

The whole process to research the microstructure and solute distribution of epitaxial growth of WAAM are illustrated in Figure 3. A macroscopic heat transfer model was built to calculate the transient temperature and solidification parameters. Then, they were input to the PF model, which takes the misorientation angle into consideration to obtain the microstructure and concentration field. Finally, metallographic examinations were conducted to validate the dendrite morphology.

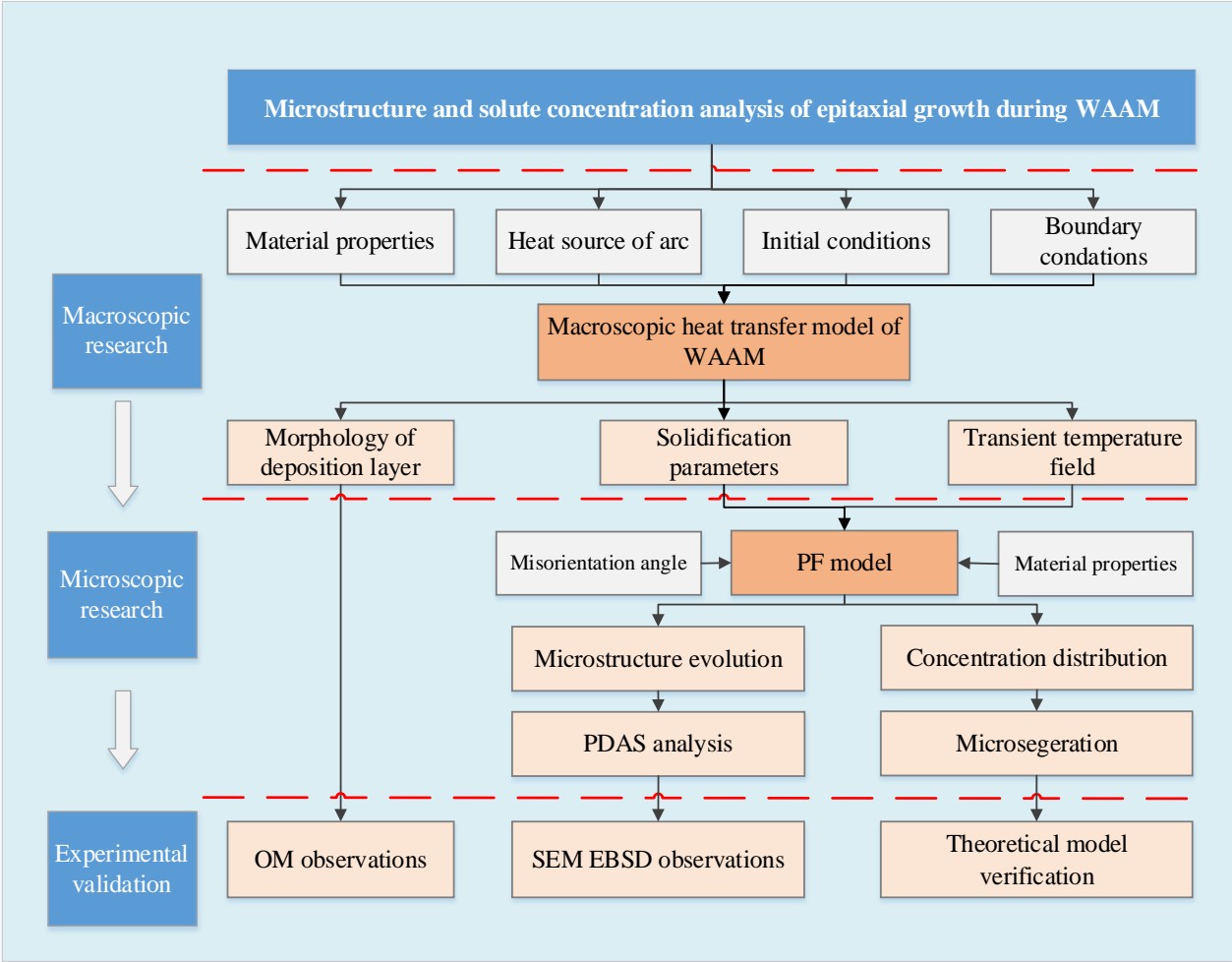

**Figure 3.** Flowchart of the multiscale model.

## 3. Results and Discussion

### 3.1. Macroscopic Temperature Distribution

Many process parameters affect the deposition process in WAAM, among which, the current of the heat source and the travel speed have a more significant influence on the deposition and solidification process. Therefore, travel speed was selected as the main variable in this paper.

Figure 4 shows the temperature distribution of the stable deposition stage for the 2319 Al alloy in the WAAM process. The temperature distributions were extracted from the FE model along the longitudinal section direction in Figure 1. The current of the heat source remained constant at 200 A, and the travel speed ranged from 3 mm/s to 5 mm/s. The travel speed is simply the speed at which the welding torch or gun is moved across the workpiece. The maximum temperature at the molten pool decreased with increasing travel speed, varying from 1059 °C to 987 °C. Meanwhile, the size of the molten pool also shrinks with increasing substrate speed. This was because, the faster the substrate moving speed, the lower the heat input in the molten pool. Consequently, both the maximum temperature and the molten pool size decreased with increasing travel speed.

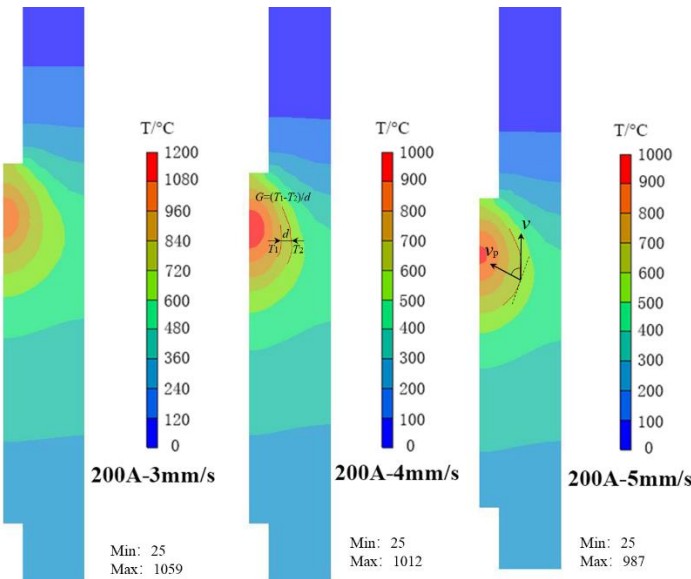

**Figure 4.** Temperature field distribution of WAAM at current and travel speeds of 200 A and 3 mm/s to 5 mm/s.

The solidification process, e.g., temperature gradient and solidification speed near the fusion line, can be obtained from the macroscopic FE model, which is the basis of calculating the microstructure evolution in PF simulations. The temperature gradient is defined as the temperature change per unit length perpendicular to the fusion line, and it was extracted in the liquid phase in front of the S/L interface. Therefore, the temperature gradients under several certain processing parameters were calculated from the macroscopic temperature distribution by the FE model. Figure 5 demonstrates the 3D contours of the temperature gradient changing with current and travel speed, illustrating that a lower temperature gradient resulted from a higher substrate moving speed and lower arc current. For the selected processing parameters of current (200 A) and travel speed (3–5 mm/s) in this work, the temperature decreased from 200 K/mm and 150 K/mm to 120 K/mm.

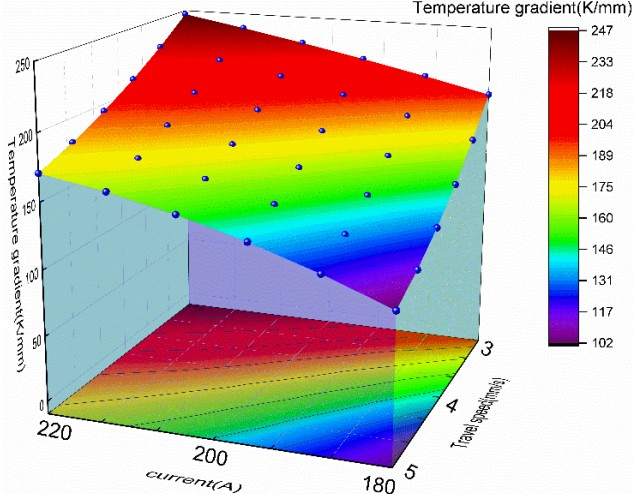

**Figure 5.** Three-dimensional temperature gradient changes with selected processing parameters.

Solidification speed is another key parameter in PF simulations. It is determined by the travel speed and the selected position where the microstructure forms in the molten pool. Therefore, the solidification speed can be obtained by [25]:

$$v_p = v \cos \alpha$$

where $v$ is the travel speed of WAAM, and $\alpha$ is the angle between the fusion line normal and the direction of $v$.

### 3.2. Microstructure Evolution with Epitaxial Growth

The microstructure evolution can be predicted by the PF model after the acquisition of solidification parameters in the WAAM deposition process. Since epitaxial growth can be observed in WAAM solidification, the dendrite growth and solute concentration distribution with different misorientation angles were simulated. Solidification starts with solid nucleating on the solid substrate at the fusion boundary, and the newly produced crystal inherits the preferential orientation from the existing grains of the substrate material. Therefore, even in a tiny domain, the dendrites can grow along different directions, namely, epitaxial growth, in the solidification process.

The calculation domain was chosen to be $400 \times 300$ uniform meshes, with each cell being the size of 0.216 μm. In the middle part of the domain, the misorientation angle was set as 25 degrees to present the epitaxial growth phenomenon. Figure 6 shows the microstructure evolution and concentration distribution process with epitaxial growth, under the processing parameters of 200 A-5 mm/s during WAAM solidification.

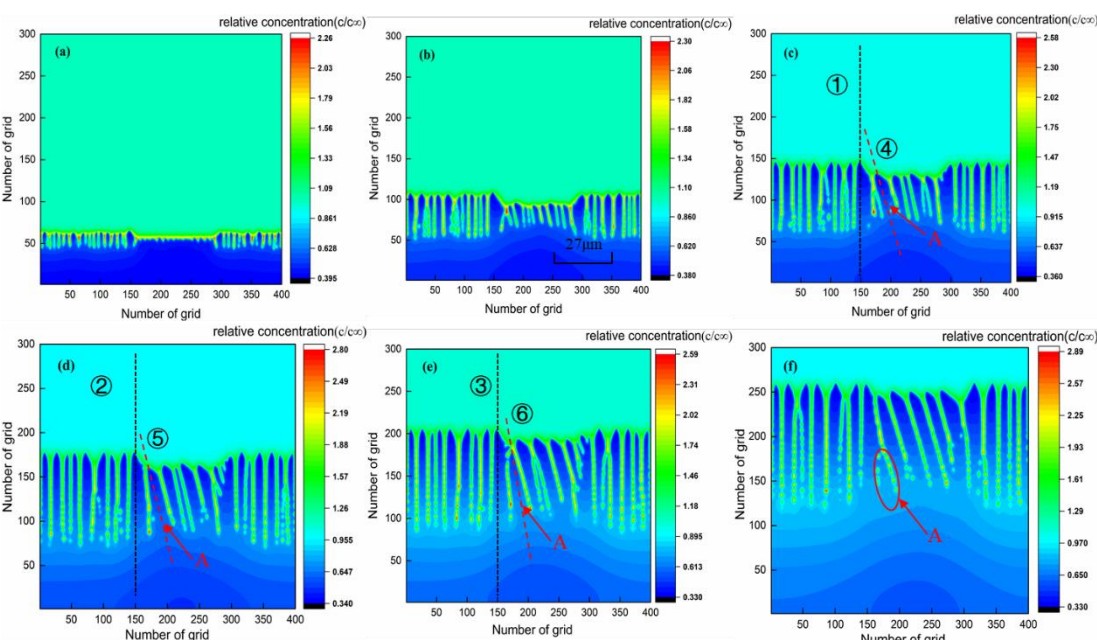

**Figure 6.** Microstructure evolution and concentration field with epitaxial growth at different times: (**a**) t = 2500Δt; (**b**) t = 4500Δt; (**c**) t = 6500Δt; (**d**) t = 8500Δt; (**e**) t = 10,500Δt; (**f**) t = 14,500Δt. Line ①,②,③ are the axis passing through the selected dendrites and line④,⑤,⑥are the axis passing through the inclined dendrites. A is the dendrite at the juncture zone of favorably oriented grains.

The interface became unstable, and some small grains appeared in the initial stage of solidification. In the middle part of the domain, where the misorientation angle was 25°, the time for interface instability, and the grains beginning to appear, occurred behind other zones [Figure 6a]. The small grains continued to grow, driven by supercooling in front of the interface. Obviously, as shown in Figure 6b, the grains with a 25-degree misorientation were much smaller than the grains on the right and left sides. As the grains grew, a few small cellular dendrites formed [Figure 6c], while the dendrites in the middle part grew along the upper left direction, not parallel to the thermal gradient. These cells went through a competitive growth stage because of their different growth direction and

limited growth space in front of the dendrite tip. Only a few dendrites survive this stage and become primary dendrite arms. As demonstrated in Figure 6c–f, dendrites A, which were at the juncture zone of favorably oriented grains and unfavorably oriented grains, were unable to survive the solidification process. This was because the dendrites with crystallographic orientations close to the parallel direction of the temperature gradient had the maximum heat flux direction, which was the most ideal growth direction. The inclined dendrites were physically hindered by the favorably oriented grains and, consequently, easily eliminated. Therefore, dendrites with misorientation angles can be observed in the WAAM solidification process, but this is not a frequently appearing phenomenon.

Not only does the competitive behavior among dendrites with various misorientation angles impact the morphology of the ultimate microstructure, but solute redistribution and the interactions between adjacent grains also influence the microstructure evolution. According to the chemical composition of the 2319 Al alloy, it is assumed to be an Al-Cu binary alloy in the PF simulations. Initially, the solute of the Al alloy (Cu element) was expelled from the newly formed solid phase, and solute enrichment appeared in front of the interface, as shown in Figure 6a. For dendrites without a misorientation angle, the solute near the dendrite tip had enough space to diffuse into the liquid far from the interface. On the other hand, the dendrites with misorientation angles grew obliquely, and were inhibited by the dendrites that grew ahead of them. Moreover, the solute concentration ahead of the inclined dendrite tip was higher than that of the dendrites perpendicular to the isotherm of the molten pool. The higher the concentration, the more difficult it is for the solute to diffuse sufficiently. This phenomenon made the inclined dendrites much easier to eliminate in the competitive stage of the solidification process.

Figure 7a shows the changing curves of the relative concentration along the axis (lines ①②③ in Figure 6a–c) of the dendrite, without a misorientation angle, at three different times in the solidification process. These three curves had almost the same pattern for the concentration distribution. The solute content was lowest in the solid phase, and then increased rapidly when it was close to the interface from the solid side. The solute concentration reached its peak in the liquid phase in front of the interface. Subsequently, the concentration gradually dropped to the initial liquid composition far from the interface. When compared with the three curves, there was an obvious difference among them: the curve with less calculation time lagged completely behind the curves with more calculation time, because the dendrite grew longer and longer with time.

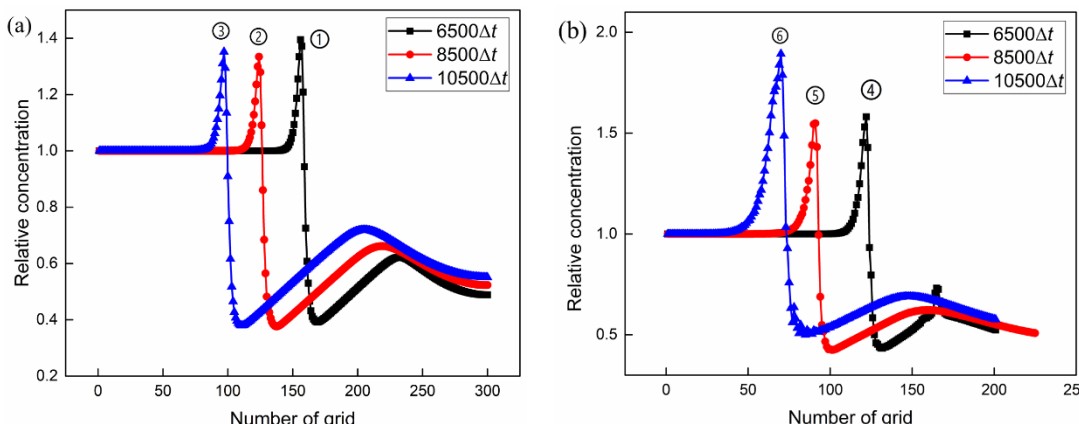

**Figure 7.** Concentration distribution along selected lines in Figure 6. (**a**) Lines passing through axis of dendrites without misorientation; (**b**) Lines passing through axis of inclined dendrites.

Figure 7b illustrates the relative concentration along the inclined dendrite axis (Line ④⑤⑥ in Figure 6a–c) at different times. With the growth of dendrites, the concentration distributions along the axis always presented a similar tendency. Unlike the concentration curves of dendrites without misorientation, whose peak values remained almost stable

at approximately 1.4, there was a sudden increase in the peak value of the concentration distribution curves from approximately 1.6 (at 6500$\Delta t$) to 1.9 (at 10,500$\Delta t$). Combined with the contours of the solute distribution in Figure 7, the selected dendrite at 10,500$\Delta t$ is surrounded closely by another two grains. Therefore, the combined action of solute enrichment of these three dendrites made the peak value much higher than that of the other two curves. Under this circumstance, the space in front of the selected dendrite tip was insufficient for the solute to diffuse to the liquid phase, obstructing the continuous solidification. Finally, the dendrite surrounded by solute enrichment and primary dendrite arms was eliminated in the subsequent solidification process.

To investigate the impacts of the travel speed on grain growth, and corresponding to the FE simulations in Figure 3, dendrite evolution at a welding current of 200 A and travel speeds of 3 mm/s, 4 mm/s, and 5 mm/s was computed, as shown in Figure 8. Within the same calculation time, the length of cellular dendrites was positively associated with the travel speed. In addition, the primary dendrite arm spacing (PDAS), which is the main parameter to quantitively evaluate the cellular dendrites, had a slightly downward trend, with increasing travel speed R. This was because the temperature gradient G was much lower at a high travel speed, and, hence, the value of GR decreased with the travel speed. According to Hunt's model [26], the PDAS increases with the value of GR, which qualitatively proves the correctness of the PF simulations in Figure 8.

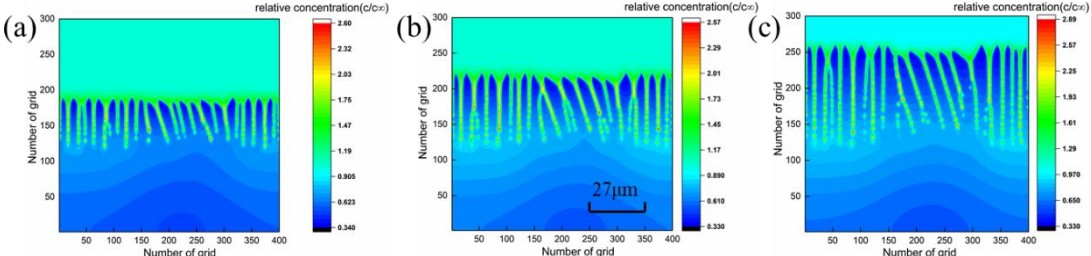

**Figure 8.** Concentration fields simulated under different process parameters. (**a**) 200 A-3 mm/s; (**b**) 200 A-4 mm/s; (**c**) 200 A-5 mm/s.

The misorientation angle is a critical parameter that affects the competitive behavior and ultimate microstructure in the solidification process. Since the misorientation angle is determined by the preferred orientation of existing base metal grains, it is essential to investigate its influences on microstructure evolution. Figure 9 demonstrates the morphology and concentration distribution of the microstructure at a current and substrate speed of 200 A-5 mm/s. The only variable parameter in this group was the misorientation, ranging from 15° to 35°.

It is obvious that the inclined angle of the dendrites increases with the misorientation angle. When the misorientation angle was 35°, the microstructure pattern was affected significantly. Some secondary dendrite arms appeared at the zone where misorientation changed from 0 to 35°. In the same zone, the competition behavior was fiercer, and fewer dendrites survived this stage. In addition, as shown in Figure 9d, the concentration in front of the interface increased gradually with the misorientation angle.

According to the principle of solute redistribution in binary alloy solidification, the concentration distribution in the initial stage of dendrite growth could be defined with the equation [27] below:

$$C_L = C_0 \left[ 1 + \frac{1-k}{k} \exp\left( -\frac{Rx}{D} \right) \right] \qquad (7)$$

where $k$ is the equilibrium partition coefficient, $D$ is the solute diffusivity in the liquid, and $R$ is the solidification speed. It could be inferred from the equation that increasing the solidification speed will result in an increase of the concentration in front of the interface. Moreover, a greater misorientation angle will lead to a smaller solidification speed. In this way, the phenomenon in Figure 9d can be verified qualitatively by Equation (7).

This means that the larger the misorientation angle, the smaller the driving force for dendrite growth. Hence, for cellular dendrites with higher misorientation angles, it is much easier to be eliminated by competitive growth. As discussed in Figure 7, the dendrites with misorientation angles lagged the dendrites growing along the temperature gradient, and, as illustrated in Figure 9a–c, the height differences between them were positively related to the misorientation angle.

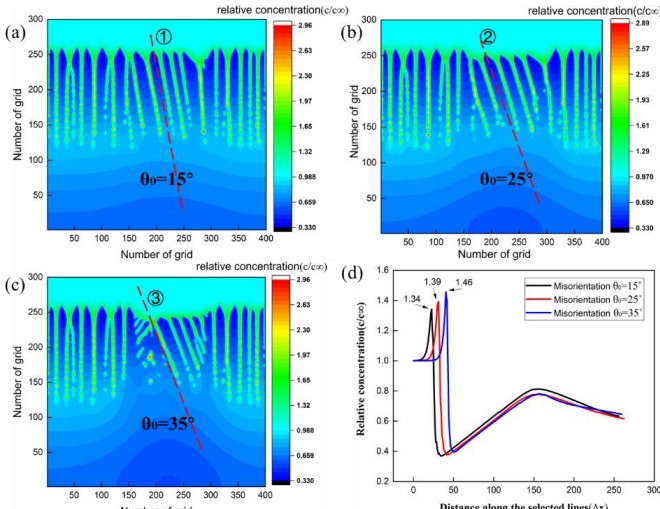

**Figure 9.** Dendrite morphology and concentration distribution at different misorientation angles. (**a**) $\theta_0 = 15°$; (**b**) $\theta_0 = 25°$; (**c**) $\theta_0 = 35°$; (**d**) concentration distribution along selected lines ①②③.

## 4. Experimental Results

Figure 10 compares the transverse cross section of the deposition layer under the process parameters of a constant current of 200 A and travel speeds of 3 mm/s, 4 mm/s, and 5 mm/s. An obvious dividing line can be seen between the substrate and deposition layer, namely, the fusion line. The width of the deposition layer decreased as the travel speed increased from 3 mm/s to 5 mm/s. Moreover, the indentation of the fusion line at a speed of 3 mm/s was rather evident, while the penetration depth decreased when the speed increased. This phenomenon proved the downward trend in the depth of the molten pool when the travel speed increased, which can validate the simulation results qualitatively. Since the FEM cannot take the fluid flow in the molten pool into consideration, the simulated results of temperature and molten pool size are a little lager than the measured data.

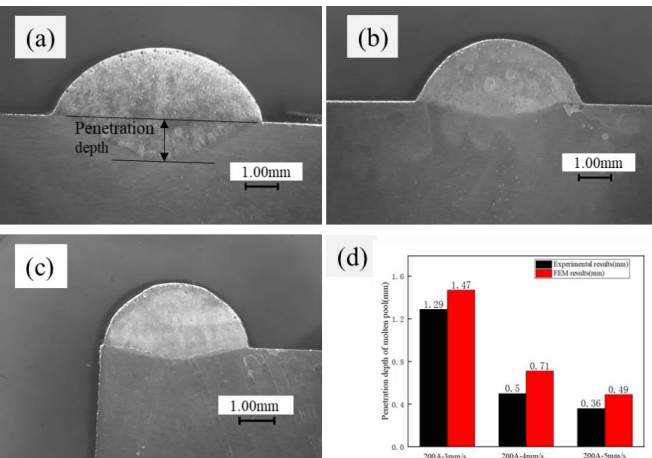

**Figure 10.** Macroscopic morphology of deposition layer from transverse cross, fabricated with welding current of 200 A and travel speeds of (**a**) 3 mm/s, (**b**) 4 mm/s, and (**c**) 5 mm/s; (**d**) comparison chart between the experimental results and the simulation results.

To validate the PF simulation results, metallographic observations were conducted on the WAAM specimen to reveal the morphology of the microstructure. Three samples were selected to observe the microstructure (Figure 11) at the bottom of the molten pool with SEM.

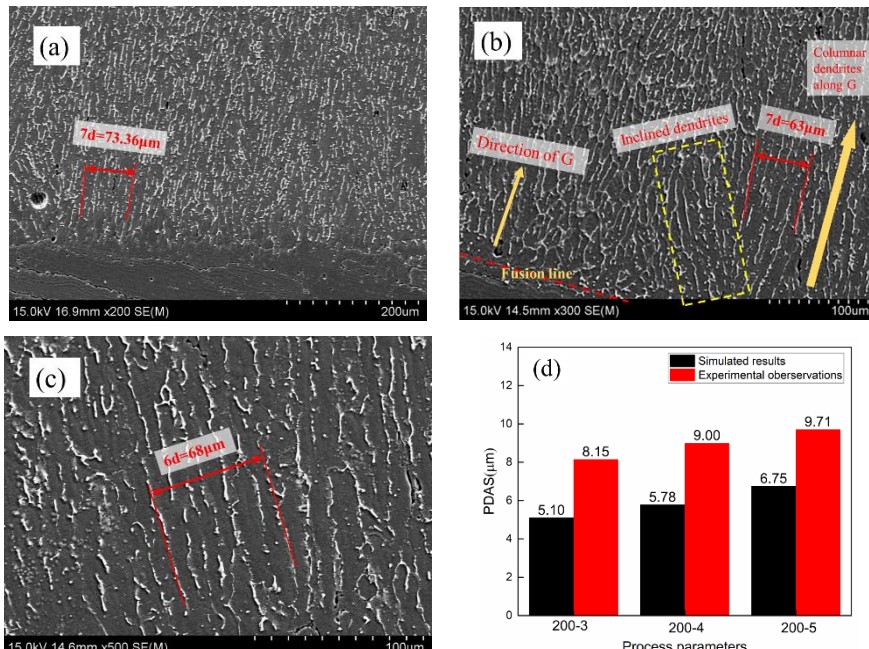

**Figure 11.** Metallographical observations of WAAM specimen under process parameters of (**a**) 200 A-3 mm/s; (**b**) 200 A-4 mm/s; and (**c**) 200 A-5 mm/s; (**d**) comparison of spacing obtained by simulation and experiment.

Figure 11a–c shows the microstructure of the 2319 Al alloy specimen fabricated by WAAM, with current and travel speeds of 200 A-3 mm/s, 200 A-4 mm/s, and 200 A-5 mm/s, respectively. The sampling position under the SEM was at the bottom of the molten pool, and the fusion line can be seen at the lower corner in Figure 11a,b. Most cellular dendrites growing along the same direction are observed in the graphs whose direction is perpendicular to the fusion line. Only a few inclined dendrites can be seen in Figure 11b. Since the grains tended to grow up along their preferred orientation (<100> direction in FCC metal) in the solidification process, if their growing direction was perpendicular to the fusion line (the heat flux is highest in this direction), then they would acquire more driving force to grow up and survive from the competition stage, which is why almost all dendrites observed under SEM were parallel to the direction of the temperature gradient. Note that, in the lower middle part of Figure 11b, some dendrites are obviously not parallel to the temperature gradient. They grew toward the upper left direction. It can be inferred that the preferential orientation of these dendrites with unique direction was inherited from the existing substrate material grains near the fusion line [28], and, after going through the competitive stage, the inclined dendrites eventually formed.

The experimental measurements of the PDAS, and the results simulated by the PF model, are shown in Figure 11d. Even though the measurement values are generally larger than the PF results, the upward trend of PDAS with travel speed has been well validated. According to Hunt's model [26], the PDAS increases as the temperature gradient decreases, meaning that the PF simulations were qualitatively consistent with Hunt's theoretical model. It is difficult to predict the dendrite arm spacing precisely, since the solidification environment in the molten pool is rather complicated, and the dendrites commonly differ with each other because of the random disturbance in the solidification process.

To prove the crystallographic orientation of grains near the fusion line, EBSD tests were conducted on the WAAM specimen, shown in Figure 12. The microstructure at the

substrate presented as coarsened equiaxial grains with random crystallographic orientation, while cellular dendrites formed at the bottom of the molten pool. It is interesting to note that some cellular dendrites near the fusion line had the same color as the grains below the fusion line. That is, in the initial stage of solidification, the substrate material was partially melted, and nucleation occurred on the base metal grains. In the subsequent solidification process, the dendrites grew and inherited the crystallographic orientation of the substrate material grains. If the preferred orientation of these dendrites is not parallel to the thermal gradient, then they undergo competition growth, and only a small part of the dendrites can survive. The growth direction of these surviving dendrites would not be perpendicular to the fusion line, as marked with a yellow box in Figure 11. Overall, the simulated results of the epitaxial growth, competitive behavior, and morphology of the microstructure were in good agreement with the experimental observations.

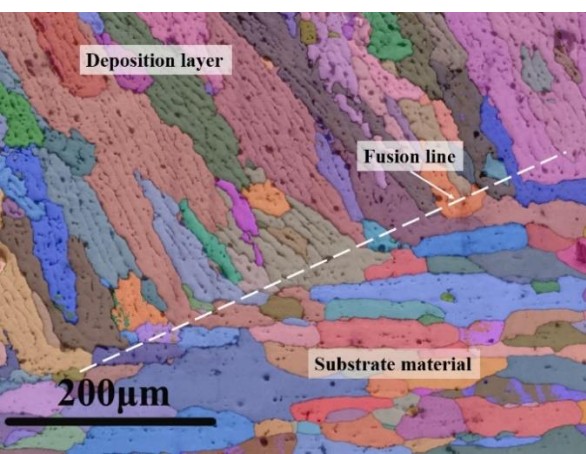

**Figure 12.** EBSD at bottom of the molten pool.

## 5. Conclusions

The microstructure evolution with epitaxial growth and the concentration distribution of 2319 Al alloy in WAAM were investigated by developing a multiscale model, combining FEM and PF models. The conclusions are as follows:

(1)  A macroscopic FE model was developed to compute the transient temperature field with different process parameters in the WAAM process. Temperature gradients in front of the S/L interface were extracted as input to the PF model to obtain the microstructure. The effects of process parameters on the temperature gradient were revealed: a lower temperature gradient results from a higher travel speed and lower arc current.

(2)  The microscopic PF model was employed to simulate the microstructure evolution and concentration distribution of the solidification process in WAAM. The dendrites with misorientation angles grew obliquely, and the competitive behavior affected by epitaxial growth is represented. When the dendrites were surrounded by other grains, the peak concentration increased suddenly, and the dendrites were inhibited by the grains growing ahead of them. Consequently, the dendrite arms would be eliminated in the subsequent solidification process.

(3)  The influence of misorientation angles on microstructure morphology and solute distributions was investigated. The inclined angle of the dendrites increased with the misorientation angle, and, similarly, the concentration in front of the interface increased gradually with the misorientation angle.

(4)  Metallographic observations were conducted on the WAAM specimen to validate the PF results. The vast majority of cellular dendrites growing along the temperature gradient direction were observed in the graphs. Only a few inclined dendrites appeared. EBSD tests showed that some cellular dendrites near the fusion line had the same

preferred orientation as the grains below the fusion line, which proved the existence of epitaxial growth.

**Author Contributions:** R.G. contributed to the study conception, design and wrote the first draft of the manuscript. Material preparation, data collection and analysis were performed by Y.C. and L.C. All authors commented on previous versions of the manuscript. All authors have read and agreed to the published version of the manuscript.

**Funding:** This work was financially supported by the National Natural Science Foundation of China (Grant No. 52205432, 52275376); Natural Science Foundation of the Higher Education Institutions of Jiangsu Province, China. (Grant No. BK20221118), China Postdoctoral Science Foundation (2022M723375).

**Data Availability Statement:** Not applicable.

**Conflicts of Interest:** The authors declare no conflict of interest.

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
