# Peer review of "Microstructure and Solute Concentration Analysis of Epitaxial Growth during Wire and Arc Additive Manufacturing of Aluminum Alloy"

_crystals, doi:10.3390/cryst13050776_

Round 1
Reviewer 1 Report
This paper reports competitive solidification microstructure evolution in a wire arc additive manufacturing for an aluminum alloy by using phase field method. Firstly, temperature gradient values were estimated by macroscopic FEM thermal analyses in several torch traveling velocities. The obtained temperature gradients and the solidification speeds were applied to the phases field analyses. The epitaxial solidification microstructure evolution was performed in two different crystallographic preferred orientations, one was the same direction of temperature gradient, and another was inclined direction. Finally, the competitive dendrite growth phenomenon was discussed from the point of solidification theory and was compared with the experimental measurement.
It seems that this paper lacks many important values and information to understand numerical methods and results.
I recommend that it be accepted for publication after major revision. Some comments are listed as follows:
1. (m) In Fig.1, the substrate sizes, length, width and thickness, must be indicated. The deposition sizes, width and height, also necessary.
2. (m) The reference numbers are not correctly inserted, showed as “Error! Reference source not found” at so many places in this article. Please fix them.
3. (m) What does mean the abbreviation “SL” in lines 124. Is it “solid and liquid”? Please put the full words.
4. (m) What is λ in Eq.(1). Please explain it.
5. (m) Is “τ is the relaxation time” in line 136 correct? “Relaxation time” is usually used in Lattice Boltzmann method. In phase field method, this τ is called as interface mobility or kinetic. please check it.
6. (m) In phase field methos parameters, W and λ are related with interfacial energy and interface thickness. Please show their values.
7. (m) It seems that c0 in line 140 and c∞ in line 141 have the same meaning, 6.3wt%. If they are different, please put more explanation.
8. (m) In phase field simulation, ER2319 alloy composition is assumed as the binary system, Al-6.3Cu. Please explain appropriate reason of the assumption.
9. (m) Please show the physical constants values, liquidus slope m, partitioning coefficient k and solute diffusivity D. The anisotropy strength γ value also be showed.
10. (m) In Fig.4, are these cross-sections in the center of Fig.1? please explain them. please write each snapshot times from 3 to 5mm/s.
11. (m) Please show the time difference Δt value of the caption in Fig.6 into around lines 25 as the numerical conditions.
12. (m) In line 271, “Fig.5” is right? It seems to be Fig.6.
13. (m) Around lines 288-293, PDAS obtained by the phase field calculations is discussed by Hunt model. Hunt model was proposed on assumption of constant interface velocity. In the present PF calculations in Fig.6, does the interface velocity reach constant value? Please discuss about this point.
Reviewer 2 Report
The paper "Microstructure and solute concentration analysis of epitaxial growth during wire and arc additive manufacturing of aluminum alloy" presents the experimental investigation and finite element simulation of the microstructure formation in the aluminum alloy during additive manufacturing. The authors have determined the dependence of the effects of process parameters on the temperature gradient. EBSD investigation also qualitatively approved the epitaxial growth obtained by the simulation. The paper may be accepted for publication. However, some points of the paper should be clarified accordingly following comments:
1. Most of the analyzed references are too old in the introduction part. It is recommended to consider the last works about microstructure formation during additive manufacturing (please, consider the works of I.S. Loginova et al).
2. The authors have determined a texture in the microstructure near the fusion line only qualitatively (by observation of similar colors). However, the EBSD technique let to determine the texture parameters of the grain microstructure quantitatively. It is recommended to use this possibility and compare the simulation and experimental results at a higher scientific level.
3. The error of the calculation of the microstructural features is too high for the 4 and 5 mm/s travel speeds. It decreases of applicability of the simulation for a wide range of process parameters. It is recommended to check the models' parameters and discuss the reason for a such large difference between experimental and calculated values of penetration depth and the primary dendrite arm spacing.
4. The mesh size for the FE simulation is too large. It may be a reason for the large error calculation described in comment #3. It is recommended to decrease the mesh size.
5. The prehistory of obtaining wire and substrate materials should be described in the Experimental setup part. The initial microstructure of the substrate may have a significant influence on the microstructure formation.
6. Minor corrections:
- Some Reference sources are not found in the text.
- A description of the experimental methods of the microstructure investigation should be given in the manuscript.
Round 2
Reviewer 1 Report
I confirmed that all commented points were precisely answered and revised.
Reviewer 2 Report
The authors have answered previous comments and improved the manuscript. The paper may be accepted for publication.